# Investigating the Linkage between Economic Growth and Environmental Sustainability in India: Do Agriculture and Trade Openness Matter?

**Ayhan Orhan** [1], **Tomiwa Sunday Adebayo** [2], **Sema Yılmaz Genç** [3] and **Dervis Kirikkaleli** [4,*]

1 Economics Department, Faculty of Economic and Administrative Science, Kocaeli University, Kocaeli 41380, Turkey; aorhan@kocaeli.edu.tr
2 Department of Business Administration, Faculty of Economics and Administrative Science, Cyprus International University, Nicosia 99258, Turkey; 21905547@student.ciu.edu.tr
3 Department of Marketing and Advertising, Ali RızaVeziroğlu Vocational School, Kocaeli University, Kocaeli 41780, Turkey; semayilmazgenc@kocaeli.edu.tr
4 Department of Banking and Finance, Faculty of Economics and Administrative Sciences, European University of Lefke, Lefke 99010, Turkey
* Correspondence: dkirikkaleli@eul.edu.tr

**Abstract:** This paper assesses the linkage between $CO_2$ emissions and economic growth while taking into account the role of energy consumption, agriculture, and trade openness in India. Using data covering the period between 1965 and 2019, the Bayer and Hanck cointegration and Gradual shift causality tests are applied to assess these economic indicators relationships'. Furthermore, we employed the wavelet coherence test. The advantage of the wavelet coherence test is that it differentiates between short-, medium-, and long-run dynamics over the entire sampling period. To the best of the authors' understanding, the present paper is the first to apply wavelet analysis to investigate this relationship by incorporating agriculture as a determinant of environmental degradation. The empirical outcomes show that all variables appear to be highly correlated with $CO_2$ emissions with the exemption of trade openness. This is further affirmed by the Gradual shift causality test, which shows that agriculture and energy consumption are crucial determinants of $CO_2$ emissions in India. Accordingly, adequate policy measures are proposed based on these findings.

**Keywords:** environmental sustainability; agriculture; economic growth; trade openness; energy consumption; India

## 1. Introduction

The most recent Sustainable Development Goals (SDG) performance document on Asia and the Pacific parties [1] reveals these nations' incompetence in dealing with the problem of rising pollution. Although developing nations are making substantial strides toward a stable energy future whilst also enhancing environmental sustainability, they are witnessing an uptick in emissions while still struggling with the problem of energy security. One main cause of these problems is the fossil fuel-based economic development trend in these countries [2]. This continued dependence on fossil fuel solutions is pushing these countries to abandon SDG 13, i.e., climate change action. Since these countries are already developing, achieving economic development has taken precedence over maintaining environmental sustainability. Regarding the growth pattern of these nations, the SDG Progress Document 2019 [3] found that nations in south and southwest Asia are lagging behind in meeting the SDG 13 goals.

Although these countries have made modest strides in meeting the SDG 8's goals of respectable employment and economic development, this growth trajectory has been con-

sidered unsustainable. This problem was illustrated in the United Nations' new study on SDG achievement [3], which addressed these countries' preference for investment in fossil fuels rather than climate-related practices. India is also extremely vulnerable to climate change, mainly due to monsoon shifts and the melting of the Himalayan glaciers. The nation has committed to a 33–35 percent reduction in its economy's "emissions intensity" by 2030, relative to 2005 levels. The primary energy mix of India in 2019 is depicted in Figure 1. Coal accounts for a significant amount of energy consumption, where pollution is a significant byproduct. In 2019, India was recognized as the third largest emitter of GHGs in the world [4]. This illustrates that economic activity and GHGs emissions are rising concurrently. Nonetheless, if the nation does not focus on curbing the unnecessary use of coal, its dream of transitioning to a low-carbon economy will be unsuccessful. At present, to maintain its economy, the nation remains dependent on fossil fuels.

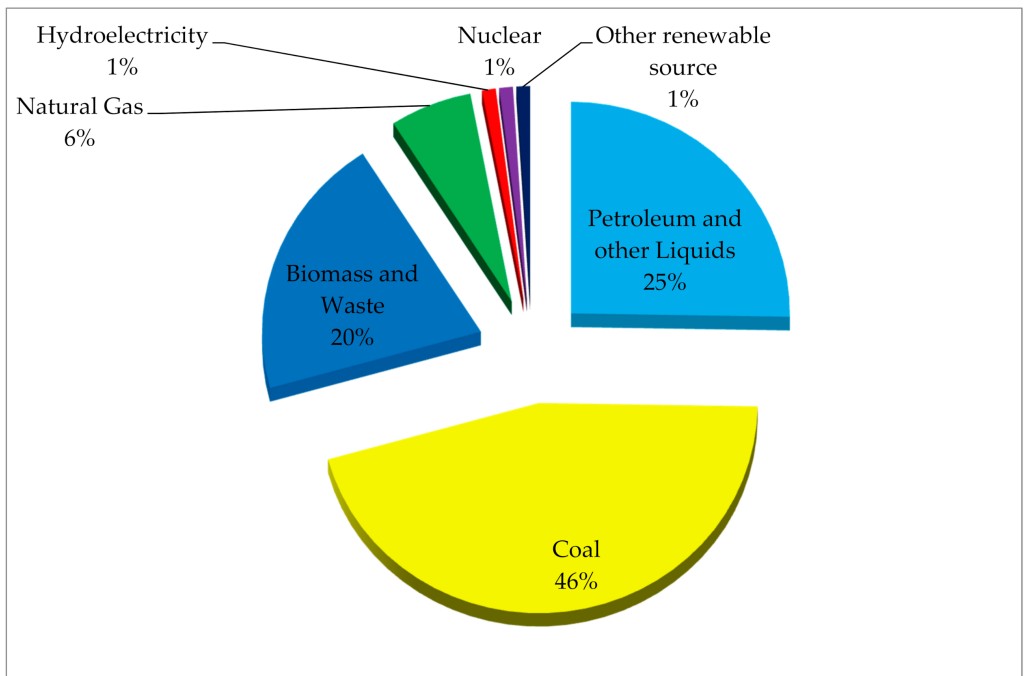

**Figure 1.** India's totalprimary energy consumption by fuel type.

This study examines the interconnection between $CO_2$ emissions and trade openness, economic growth, energy consumption, and agriculture. It is important to note that the policy process can be structured in such a manner that SDG 7, SDG 13, SDG 8, and SDG 12 will all be discussed. Energy consumption, agriculture, economic development, and trade openness patterns can all be taken under one policy umbrella in this way. In line with the UNESCAP [1] and ADB [5], it is clear that India is having difficulties in achieving sustainable growth as a result of its current economic and related policies.

This research is distinctive from prior studies [6–11], which analyzed this association using time domain analyses such as the autoregressive distributed lag (ARDL), vector error correction model (VECM), fully modified ordinary least square (FMOLS), dynamic ordinary least square (DOLS), ordinary least square (OLS), and general method of moments (GMM) to investigate the impacts of agriculture, economic growth, trade openness and energy consumption on $CO_2$ emissions. In the economic literature, time-domain analysis is the most widely used method for studying time series. Individual parameter evolution is constructed and multivariate associations are measured over time using this method. Another body of research has concentrated on frequency-domain analysis. In the context where all time and frequency domains are taken into account, the wavelet approach (WA) reconciles both approaches. Using this tool, the approach differentiates between short-, medium-, and long-run dynamics over the entire sampling duration. The wavelet

transformation is an effective method for signal analysis and processing that is incredibly useful in a variety of areas, including denoising and compression, and working with nonstationary signals as images. Long-term dynamics at low frequencies (backgrounds) are referred to as patterns, whereas short-term dynamics at high frequencies (discontinuity, edges) are referred to as anomalies. Although the latter encompasses a small portion of the image, they contain multiple details and must be properly depicted.

As stated by [12,13] there are several fascinating features associated with the wavelet transform: (i) because of its strong time-frequency localization capabilities, it can analyze signals with features that change over time; (ii) it gives a depiction on various scales (multiresolution representation); and (iii) it can be achieved via a filter bank. In the literature, several papers have assessed the impact of agriculture, energy use, economic growth, and trade openness on environmental sustainability. However, to the best of our knowledge, the present paper is the first to apply wavelet analysis to investigate this analysis by incorporating agriculture as a determinant of environmental sustainability into the model.

The remainder of this research is compiled as follows: the empirical and theoretical framework is depicted in Section 2. The data and methodology are illustrated in Section 3. The data analysis and discussion are portrayed in Section 4, and the conclusion is presented in Section 5.

## 2. Literature Review

This section of the research will be divided into two parts, namely the empirical review and theoretical framework. The empirical review discusses the relationship between $CO_2$ emissions and the independent variables (agriculture, energy consumption, trade openness, and economic growth). The theoretical framework of the study discusses the environmental Kuznets curve (EKC) theory.

### 2.1. Empirical Review

As previously mentioned, this section of the paper discusses prior studies regarding the interrelationship between $CO_2$ emissions and the regressors (agriculture, energy consumption, trade openness, and economic growth).

2.1.1. Synopsis of Studies between Environmental Degradation and Economic Growth

Prior scholars have assessed the discourse on the linkage between $CO_2$ emissions, which is a proxy of environmental sustainability and economic growth. Nonetheless, their findings are mixed. For instance, Zhang [14] in Malaysia, using the novel wavelet and Gradual shift causality, uncovered that real GDP exerts a positive impact on $CO_2$ emissions, which infers that an upsurge in GDP will lead to a decrease in environmental sustainability in Malaysia. Likewise, a study on the interconnection between real GDP and $CO_2$ in India using data from period 1992 to 2015 [6] unravelled that an upsurge in GDP leads to a decrease in environmental sustainability. In addition, there is evidence of one-way causality from GDP to $CO_2$ emissions, which implies that GDP can predict significant variation in environmental sustainability in India. Contrarily, using the MINT nations and utilizing the PMG-ARDL, Ahmed [15] uncovered a significant link between GDP and $CO_2$ emissions. Moreover, Adams [16], in countries with high geopolitical risk disclosed that real growth decreases environmental sustainability, while the Dumitrescu and Hurlin (DH) causality test shows feedback causality between GDP and $CO_2$ emissions. Using seven OECD countries, [17] assessed the linkage between $CO_2$ emissions and GDP. The investigators applied PMG-ARDL and D-H causality to examine this association. The findings disclosed that economic growth exerts a positive impact on $CO_2$ emissions, which implies that an economic expansion leads to a decrease in environmental sustainability. The D-H causality test also discloses a one-way causal linkage from GDP to $CO_2$. The study of [18] in BRICS nations also revealed a positive association between $CO_2$ and economic expansion. The positive interconnection between $CO_2$ and economic expansion is also

validated by the studies of [19] for Indonesia, [20] for Pakistan, [21] for Turkey, and [22] for global economy.

### 2.1.2. Synopsis of Studies between Environmental Degradation and Energy Consumption

Energy consumption is regarded as essential for economic expansion, decreasing environmental sustainability from renewable sources [13]. The study of [23] in Mexico uncovered that energy use deteriorates the quality of the environment. The frequency-domain causality test also revealed one-way causality from energy use to consumption-based carbon emissions in the short-, medium-, and longterm. In Thailand, the research of [8], using data from the period 1970–2016, disclosed that energy use exerts a positive and significant impact on $CO_2$ emissions, decreasing environmental sustainability. The outcomes of wavelet coherence also show an in-phase association between $CO_2$ emissions and energy use in Thailand. Using 12 MENA countries, the study revealed one-way causal interconnection from energy use to $CO_2$ emissions. Odugbesan and Rjoub et al. [11] assessed the interconnection between energy use and $CO_2$ emissions in Turkey using data from the period 1960–2018. The investigators applied the FMOLS, and DOLS and the findings showed that energy-use impact $CO_2$ emissions positively in Turkey. The study of Cheikh et al. [24] and Akinsola and Adebayo [25] disclosed that there is positive and significant comovement between energy use and $CO_2$ emissions, which illustrates that a decrease in environmental sustainability accompanies an increase in energy use. Likewise, the study of [7] also established positive interconnection between energy consumption and $CO_2$ emissions. The positive linkage between $CO_2$ emissions and energy use is also validated by the studies of [26] for ASEAN-5 [27] for South Asia and Adebayo [28] for Mexico.

### 2.1.3. Synopsis of Studies between Environmental Degradation and Trade Openness

Over the years, numerous scholars have assessed the linkage between trade openness and environmental sustainability. Nonetheless, their findings are mixed. In South Africa, [29] examined the link between $CO_2$ emissions and trade using data spanning between 1965 and 2008. The authors utilized the ARDL approach, and findings show that trade openness exerts a negative influence on $CO_2$ emissions in South Africa, which implies that an increase in trade openness enhances environmental sustainability. Contrarily, the study of [30] in Tunisia uncovered that trade openness exerts a positive impact on $CO_2$ emissions, which infers that a decrease in environmental sustainability accompanies an increase in trade openness. Further, by using the Granger causality test, [31] assessed the linkage between trade openness and $CO_2$ emissions using data between 1971 and 2007. The empirical outcomes revealed no evidence of causal linkage between trade openness and $CO_2$ emissions in the newly industrialized countries. The studies reported in [32] and [33] provide mixed findings on the interconnection between trade openness and $CO_2$ emissions. Using data from 1963 to 2013, Mutascu [34] assessed the impact of trade openness and $CO_2$ emissions. The study utilized wavelet tools–wavelet coherence, multiple wavelet coherence, and partial wavelet coherence to analyze this interconnection. The outcomes from this study disclosed insignificant comovement between $CO_2$ emissions and trade openness. The study of [35] for BRICS and [36] for Turkey also validated the positive association between $CO_2$ emissions and trade openness.

### 2.1.4. Synopsis of Studies between Environmental Degradation and Agriculture

Agriculture is also essential for economic growth, which also contributes to a decrease in environmental sustainability if it is not ecofriendly. The study of [37] on the influence of agriculture on $CO_2$ emissions in E7 countries between 1990 and 2014 disclosed that agriculture exerts a positive impact on $CO_2$ emissions, which infers that increase in agriculture results in a decrease in environmental sustainability. Likewise, [38] examined the association between agriculture and $CO_2$ emissions in China using data from 2004 to 2017. The investigators utilized OLS, DOLS, and FMOLS to assess this association and the out-

comes show that agriculture decreases environmental sustainability. Doğan [39] assessed the impact of agriculture on $CO_2$ emissions in China using data from 1971 to 2010. The author applied the ARDL, FMOLS, DOLS, and CCR to investigate this association, and the findings show that agriculture decreases environmental sustainability. In addition, there is evidence of one-way causality from agriculture to $CO_2$ emissions. Recently, Ref. [40] assessed the $CO_2$ and agriculture association in West African economies between 1990 and 2015 using recent panel techniques. The empirical outcomes show that agriculture impacts $CO_2$ emissions, which infers that agriculture decreases environmental sustainability. The positive linkage between $CO_2$ emissions and agriculture is validated by the study of [41] for Brazil, [42] for Pakistan, and [43] for Pakistan. Contrarily, the research of [44] on the linkage between agriculture and $CO_2$ emissions in North Africa countries using Panel FMOLS and Granger causality revealed that agriculture enhances environmental sustainability. In addition, there is evidence of unidirectional causality from agriculture to $CO_2$ emissions. Table 1 illustrates a synopsis of related studies.

**Table 1.** Synopsis of related studies.

| CO₂ Emissions and Economic Growth | | | | |
|---|---|---|---|---|
| **Author(s)** | **Period** | **Country(s)** | **Techniques** | **Conclusion** |
| Zhang et al. [1] | 1970–2018 | Malaysia | Wavelet Coherence, ARDL, Gradual Shift | GDP ⇨ $CO_2$ (+) <br> GDP ⇨ $CO_2$ |
| Adedoyin et al. [18] | 1995–2015 | Top ten earners | FMOLS, DOLS, D-H Causality | GDP ⇨ $CO_2$ (−) <br> $CO_2$ ⇔ GDP |
| Adebayo [19] | 1971–2016 | Indonesia | FMOLS, DOLS, ARDL | GDP ⇨ $CO_2$ (+) |
| Ahmed et al. [15] | 1990–2018 | Chile | NARDL | GDP ⇨ $CO_2$ |
| Kirikkaleli and Adebayo [6] | 1992–2015 | India | FMOLS, DOLS, Frequency Domain Causality | GDP ⇨ $CO_2$ (+) <br> GDP ⇨ $CO_2$ |
| Adedoyin et al. [45] | 1990–2014 | BRICS | PMG-ARDL | GDP ⇨ $CO_2$ (+) <br> $CO_2$ ⇨ GDP |
| Adams et al. [16] | 1996–2017 | Countries with high geopolitical risk | PMG-ARDL, D-H Causality | GDP ⇨ $CO_2$ (+) <br> $CO_2$ ⇔ GDP |
| Ahmad et al. [46] | 1990–2014 | OECD economies | FMOLS | $GDP^2$ ⇨ $CO_2$ (−) <br> GDP ⇨ $CO_2$ (+) |
| Khan et al. [17] | 1990–2018 | Seven OECD countries | PMG-ARDL, D-H Causality | GDP ⇨ $CO_2$ (+) <br> GDP ⇨ $CO_2$ |
| Malik et al. [20] | 1971–2014 | Pakistan | Granger Causality | GDP ⇨ $CO_2$ (+) <br> $CO_2$ ⇔ GDP |
| Kirikkaleli and Adebayo [22] | 1980–2016 | Global Economy | FMOLS, DOLS, Frequency Domain Causality | GDP ⇨ $CO_2$ (+) <br> GDP ⇨ $CO_2$ |
| Rjoub et al. [21] | 1960–2018 | Turkey | FMOLS, DOLS | GDP ⇨ $CO_2$ (+) |
| CO₂ Emissions and Energy Consumption | | | | |
| He et al. [23] | 1990–2018 | Mexico | ARDL, FMOLS, DOLS, Frequency Domain Causality | EC ⇨ $CO_2$ (+) <br> EC ⇨ $CO_2$ |
| Zhang and Zhang [47] | 2000–2017 | 30 Chinese provinces | VECM | EC ⇨ $CO_2$ |
| Adebayo [28] | 1970–2016 | Mexico | ARDL, FMOLS, DOLS, Wavelet Coherence | EC ⇨ $CO_2$ (+) <br> EC ⇨ $CO_2$ |
| Olanrewaju et al. [8] | 1970–2016 | Thailand | ARDL, FMOLS, DOLS, Wavelet Coherence | EC ⇨ $CO_2$ (+) <br> EC ⇨ $CO_2$ |
| Siddique et al. [27] | 1983–2013 | South Asia | Panel Granger Causality | EC ⇨ $CO_2$ |
| Akinsola and Adebayo [25] | 1970–2016 | Thailand | Wavelet Coherence, Granger Causality | EC ⇨ $CO_2$ (+) <br> EC ⇨ $CO_2$ |
| Cheikh et al. [36] | 1980–2015 | 12 MENA countries | PSTR | EC ⇨ $CO_2$ |
| Khan et al. [48] | 1965–2015 | Pakistan | ARDL | EC ⇨ $CO_2$ (+) |
| Odugbesan and Rjoub [11] | 1993–2017 | MINT | ARDL, Granger Causality | EC ⇨ $CO_2$ |
| Munir et al. [26] | 1980–2016 | ASEAN-5 | FMOLS, Granger Causality | EC ⇨ $CO_2$ |

**Table 1.** *Cont.*

| CO$_2$ Emissions and Economic Growth | | | | |
|---|---|---|---|---|
| **Author(s)** | **Period** | **Country(s)** | **Techniques** | **Conclusion** |
| **CO$_2$ Emissions and Agriculture** | | | | |
| Wang et al. [38] | 2004–2017 | China | GMM | AGRIC ⇨ CO$_2$ (+) |
| Aydoğan and Vardar [37] | 1990–2014 | E7 countries | OLS, DOLS, FMOLS | AGRIC ⇨ CO$_2$ (+) |
| Jebli and Youssef [44] | 1980–2011 | North Africa countries | Granger Causality | AGRIC ⇨ CO$_2$ (−) AGRIC ⇔ CO$_2$ |
| Doğan 38] | 1971–2010 | China | ARDL, FMOLS, DOLS, CCR | AGRIC ⇨ CO$_2$ ((+) AGRIC ⇔ CO$_2$ |
| Nwaka et al. [40] | 1990–2015 | West African economies | Panel Techniques | AGRIC ⇨ CO$_2$ (+) |
| Rehman et al. [42] | 1987–2017 | Pakistan | ARDL | AGRIC ⇨ CO$_2$ (+) |
| Ben Jebli and Ben Youssef [41] | 1980–2013. | Brazil | ARDL | AGRIC ⇨ CO$_2$ (+) |
| **CO$_2$ Emissions and Trade Openness** | | | | |
| Shahbaz et al. [29] | 1965–2008 | South Africa | ARDL | TO ⇨ CO$_2$ (−) |
| Mutascu [34] | 1960–2013 | France | Wavelet Coherence | TO ≠ CO$_2$ |
| Sebri and Ben-Salha [35] | 1971–2010 | BRICS | VECM | TO ⇨ CO$_2$ (+) |
| Mahmood et al. [30] | 1971–2011 | Tunisia | ARDL | TO ⇨ CO$_2$ (+) |
| Hossain [31] | 1971–2007 | Newly industrialized countries | Granger Causality | TO ≠ CO$_2$ |
| Dauda et al. [33] | 1990–2016 | 9 African nations | GMM | Mixed Findings |
| Sun et al. [32] | 1991–2014 | Several Nations | VECM | Mixed Findings |
| Cetin et al. [36] | 1960–2013 | Turkey | VECM | TO ⇨ CO$_2$ |

Note: ⇨ (+): positive relationship, ⇨ (−): negative relationship, TO: trade openness, GDP: economic growth, AGRIC: agriculture, CO$_2$: carbon emissions, EC: energy consumption, ⇨: unidirectional causality, ⇔: bidirectional causality.

### 2.2. Theoretical Foundation

The theoretical background of this study is anchored on the Environmental Kuznets Curve (EKC). This theory was propounded by Kuznets [49] based on this studying of income inequality and is called the Kuznets curve. He studied the incremental pattern of per capita income and inequality. A turning point exists along the curve, which indicates where the per capita income of rural farmers who abandon their farming activities to take up white collar jobs in urban cities eventually increases and this closes the wide gap that exists between the poor and the rich. At this point, it is expected that the income inequality gap is reduced, thus improving the per capita income of the poor farmers. After the successful application of this hypothesis by Kuznets [49], environmental economists [50,51] applied the Kuznets curve to investigate the relationship between environmental sustainability and economic growth. According to them, economic growth occurs in 3 stages: scale, structural and composite effects. In the initial stage of growth, the environment suffers until a certain point is reached (turning point); at this point, the economic growth will impact the environment positively because of the development innovations and increased environmental awareness that occurs at this stage. The initial stage is called the scale effect stage, while the turning point and the time after the turning point are called structural and composite effect stages, respectively. The scale effect stage is associated with developing economies where productive activities and economic performance are supported by non-renewable energy sources, while the last two stages are associated with developed countries where service and technological innovations dominate the economic performance. In this, study, it is expected that Indian economic growth will be achieved to the detriment of the environment and will suggest policies that will encourage the sustainable and balanced development of economic growth and the environment.

### 3. Data and Methodology

#### 3.1. Data

The present paper assesses the effect of agriculture, energy consumption, trade openness, and economic growth on CO$_2$ emissions in India, utilizing data from 1965 to 2019

for all indicators. The data description, source, and unit of measurement are depicted in Table 2. Furthermore, all the variables of interest are transformed to their natural log. This is done to ensure data conform to a normal distribution [21,52]. The flow of analysis is depicted in Figure 2 and the trend of indicators used in this study is illustrated in Figure 3a–e. The study functional form is depicted in Equation (1):

$$CO_2 = f\,(GDP,\ EC,\ TO,\ AGRIC) \tag{1}$$

**Table 2.** Variables units and sources.

| Variable | Description | Units | Sources |
|---|---|---|---|
| **GDP** | Economic Growth | GDP per capita in constant USD, 2010 | WDI |
| **TO** | Trade Openness | Trade % of GDP | WDI |
| AGRIC | Agriculture | Agriculture, fishing, and forestry, value-added | WDI |
| $CO_2$ | $CO_2$ Emissions | Per capita emissions | BP |
| **EC** | Energy Use | Energy consumption per capita (kWh) | BP |

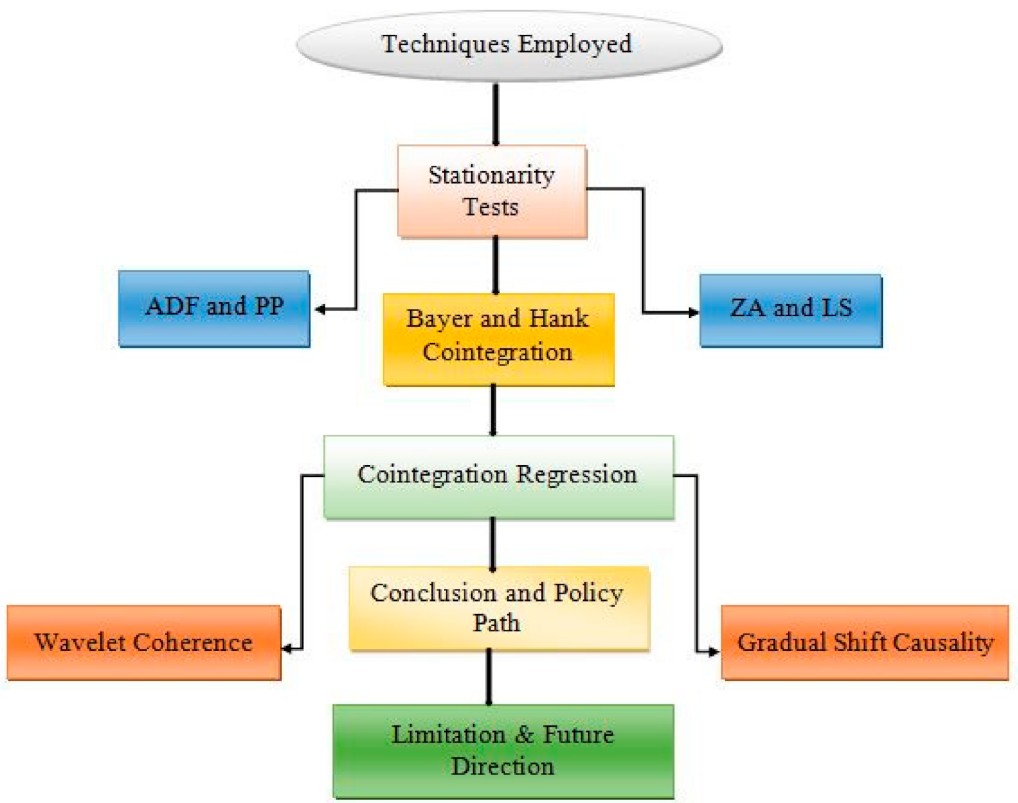

**Figure 2.** Analysis flow chart.

In Equation (1), $CO_2$ stands for carbon emissions, GDP represents economic growth, EC is energy consumption, TO illustrate trade openness, and AGRIC signifies agriculture.

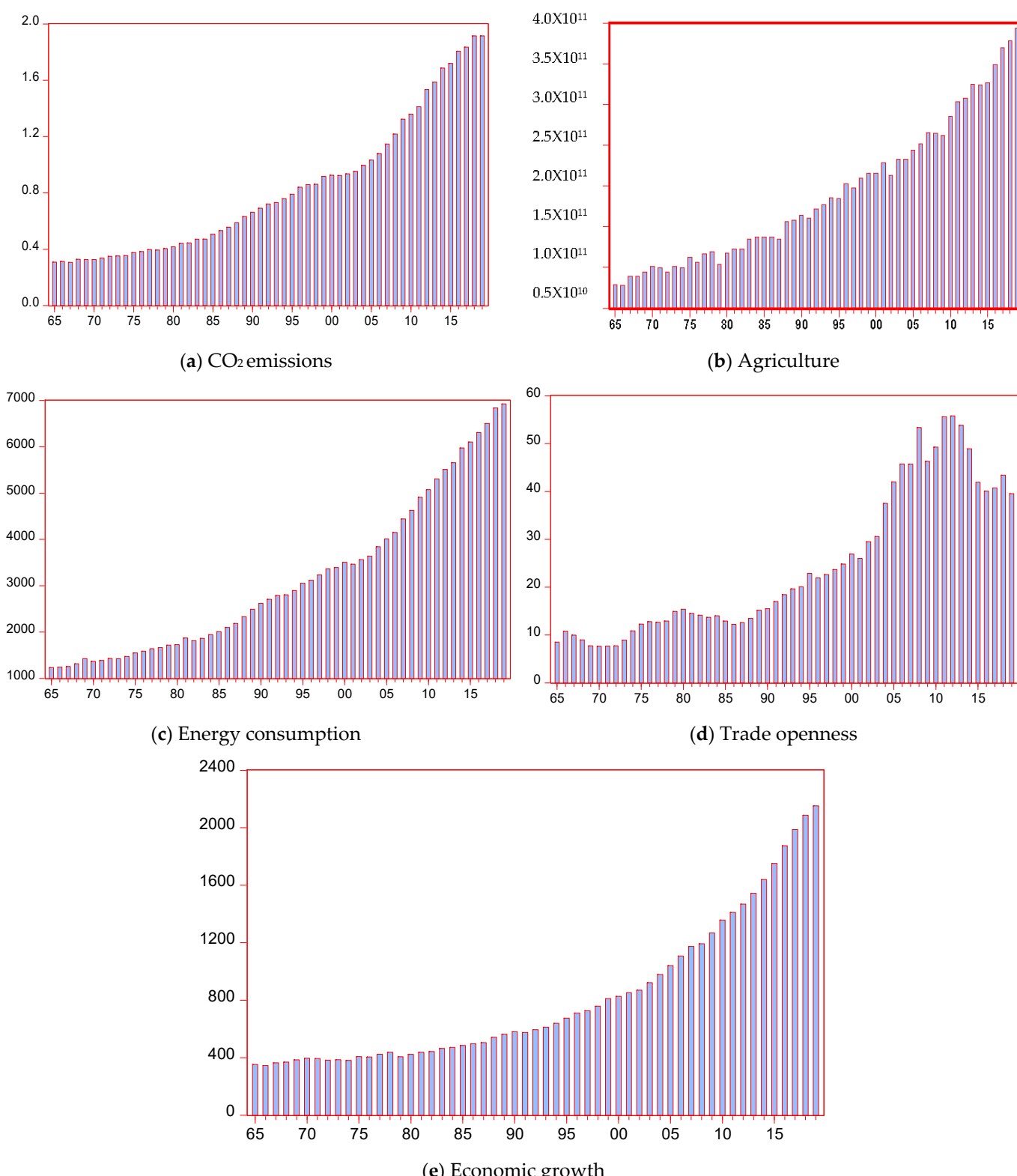

**Figure 3.** Trend of (**a**) CO₂ emissions, (**b**) agriculture, (**c**) energy consumption, (**d**) trade openness, and (**e**) economic growth.

*3.2. Methodology*

3.2.1. Stationarity Tests

Stationarity testing is important in this empirical analysis to avoid the issue of erroneous analysis. Econometric literature has a number of unit root test methods, including

KPSS proposed by [53], augmented Dickey–Fuller (ADF) suggested by [54], and PP initiated by [55]. Nevertheless, all of the tests referred to above do not account for break(s) in series, which are known to affect economic indicators. As stated by [56], if there is proof of a break in parameter, the aforementioned unit root tests (ADF, PP, KPSS, and ER) can provide biased estimates. Therefore, we employed the Zivot and Andrews's unit root test initiated by Zivot and Andrews [57]. The null and alternatives hypothesis of the ZA unit root test states unit root ($H_0$: $\theta = 0$) and no unit root (H1: $\theta < 0$). Failure to reject $H_0$ therefore means the existence of unit roots, whereas rejection is a sign of stationarity. The only drawback of the ZA root test is that it can only catch one break in series. Therefore, the unit root test [58] was included in the analysis. The benefit of LS is that it can capture both two breaks and stationarity characteristics of variable. The null and alternatives hypothesis of the LS unit root test states unit root ($H_0$: $\theta = 0$) and no unit root ($H_1$: $\theta < 0$). There is proof of unit root if $H_0$ is not rejected whereas rejection is a sign of stationarity.

### 3.2.2. Cointegration Test

It is vital to catch the long-run interconnection between GDP growth and its determinants (urbanization, energy consumption, and $CO_2$ emissions). Therefore, this study utilized the combined cointegration of [59–62]. According to [63], the needless extensive testing methods produced by other cointegration tests are eliminated by the [63] cointegration test. Furthermore, the Fisher formula is utilized in the construction of the [64] cointegration test. Equations (2) and (3) portray the cointegration [64]:

$$EG - JOH = -2[\ln (PEG) + \ln(PJOH)] \tag{2}$$

$$EG - JOH - BO - BD = -2[\ln(PEG) + \ln(PJOH) + \ln(PBO) + \ln(PBDM)] \tag{3}$$

where PEG portrays the significance level for [60], and the level of significance for Johansen [59] is portrayed by PJOH. PBDM and PBO illustrate the level of significance for the cointegration tests of [61] and [62], respectively.

### 3.2.3. Wavelet Coherence Test

The present research utilized the novel wavelet coherence test to assess the time-frequency dependence of carbon emissions ($CO_2$), and agriculture (AGRIC), energy consumption (EC), trade openness (TO), and economic growth (GDP) in India. With a wavelet analysis, a time series could be separated into frequency elements. Although the Fourier analysis has a full ability of representation and decomposition of stationary time-series, the research could be conducted with a nonstationary time-series through wavelets. Furthermore, wavelets promote the conservation of time for localized information, enabling comovement to be measured in time–frequency space. Wavelet coherence analysis is mainly time series analysis. The cross wavelet transform is defined by two stock index time series x(t) and y(t) with the continuous transforms of wx(u,s) and wy(u,s), where u is the position index, s is the scale, and* depicts the complex conjugate. Finally, to test the coherence of the cross wavelet transform in time–frequency space, and following [65,66], we apply the wavelet squared coherence called wavelet coherence, which can be defined as:

$$R^2(s) = \frac{\left| S\left(s^{-1} w_t^{xy}(s)\right) \right|^2}{S(s^{-1}|w_t^x(s)|^2) S\left(s^{-1}\left|w_t^y(s)\right|^2\right)} \tag{4}$$

The wavelet coherence can be interpreted as a correlation coefficient with a value range between 0 and 1, s denotes the smoothing parameter. In the no-smoothing case, the wavelet coherence will be equal to 1. The squared wavelet coherence coefficient varies from $0 \leq R^2(k,f) \leq 1$, with values close to 0, suggesting poor correlation and values close to 1, confirming strong correlation. As a consequence, wavelet coherence can be regarded as a

valuable method for evaluating the association of chosen parameters over time. Following Torrence and Gilbert, [67], we applied the smoothing operator Sas:

$$S(W) = S_{scale}(S_{time}(W_n(S)))　　　　　　　　　(5)$$

Smoothing along the wavelet scale axis is denoted by $S_{scale}$, and smoothing in time is denoted by $S_{time}$. It is only normal to build the smoothing operator to have a footprint identical to the wavelet in use. Torrence and Webster [65] proposed a fitting smoothing operator for the Morlet wavelet:

$$S_{time}(W)s = \left( W_n(s) \ast \frac{-1^2}{x_1^{2s^2}} \right) S　　　　　　　　(6)$$

$$S_{time}(W)s = W_n(s) \ast x_2 \Pi(0.6s)n　　　　　　　　(7)$$

where $S_{time}$ represents time smoothing, frequency (bandwidth) is depicted by $W$, normalization constants are represented by $x_1$ and $x_2$, and rectangle function is depicted by $\Pi$. In addition, dimensionless time is represented by $n$. The scale decorrelation length for the Morlet wavelet has been empirically calculated at 0.6 [67]. Both convolutions are implemented discretely in practice, so the normalization coefficients are measured numerically.

### 3.2.4. Gradual Shift Causality Test

Subsequently, this wavelet methodology is followed by the Gradual shift causality test. Toda and Yamamoto [68] established a framework, which is anchored on vector autoregression (VAR) built by Sims [69]. In calculating for the optimal lag length, p + d$_{max}$ is added to the lag of d$_{max}$, which is ascertained by the series maximum order of integration in the VAR framework. However, ignoring the structural shifts can cause the VAR model to be unreliable and contradictory [70]. For this reason, to examine the causal linkage between $CO_2$, GDP, AGRIC, TO, and EC, Nazlioglu et al. [71] developed the Fourier–TY causality test, which captures the structural shifts in Granger causality analysis and includes the gradual and smooth shift. It can also be called the "Gradualshift causality test". The Fourier Granger causality test was developed using single-frequency (SF) and cumulative frequencies (CF), respectively, known as Fourier approximation. The modified Wald test statistic (MWALT) is generated by adding the TY-VAR analysis and Fourier approximation. Assuming the coefficients of the intercept are constant over time, this modifies the VAR model into Equation (8):

$$y_t = \sigma(t) + \beta_1 y_{t-1} + \cdots + \beta_{p+dmax} y_{t-(p+dmax)} + \varepsilon_t　　　　(8)$$

where $y_t$ denotes $CO_2$, GDP, AGRIC, TO, and EC; $\sigma$ denotes intercept; $\beta$ denotes coefficient matrices; $\varepsilon$ denotes the error term; and t denotes time function. To capture the structural change, the Fourier expansion is introduced and explained, as in Equation (9).

$$\sigma(t) = \sigma_0 + \sum_{k=1}^{n} \gamma_{1k} sin\left(\frac{2\pi kt}{T}\right) + \sum_{k=1}^{n} \gamma_{2k} cos\left(\frac{2\pi kt}{T}\right)　　　　(9)$$

where $\gamma_{1k}$ and $\gamma_{2k}$ measure the frequency amplitude and displacement, respectively, andn denotes the frequency number. The structural shift is thereby considered, which defines the Fourier Toda–Yamamoto causality with cumulative frequencies (CF), as in Equation (10).

$$y_t = \sigma_0 + \sum_{k=1}^{n} \gamma_{1k} sin\left(\frac{2\pi kt}{T}\right) + \sum_{k=1}^{n} \gamma_{2k} cos\left(\frac{2\pi kt}{T}\right) + \beta_1 y_{t-1} + \ldots + \beta_{p+dmax} y_{t-(p+dmax)} + \varepsilon_t　　(10)$$

where *k* denotes the approximation frequency. The single-frequency component is defined in Equation (11):

$$\sigma(t) = \sigma_0 + \gamma_1 sin\left(\frac{2\pi kt}{T}\right) + \gamma_2 cos\left(\frac{2\pi kt}{T}\right) \tag{11}$$

The Fourier Toda–Yamamoto causality with single frequencies (SF) is defined by Equation (12):

$$y_t = \sigma_0 + \gamma_1 sin\left(\frac{2\pi kt}{T}\right) + \gamma_2 cos\left(\frac{2\pi kt}{T}\right) + \beta_1 y_{t-1} + \ldots + \beta_{p+d} y_{t-(p+d)} + \varepsilon_t \tag{12}$$

Here, the testing of the null hypothesis of noncausality is zero ($H_0$: $\beta_1 = \beta_\theta$); the Wald statistic can be used for testing the hypothesis.

## 4. Findings and Discussion

The descriptive summary of the current study's data is depicted in Table 3. The maximum and minimum values revealed that $CO_2$ ranges from 0.307033 to 1.915750, EC ranges from 1234.199 to 6923.931, and GDP ranges from 345.4216 to 2151.726, TO ranges from 7.661769 to 55.79372, and AGRIC ranges from $7.75 \times 10^{10}$ to $3.94 \times 10^{11}$. Furthermore, the Jarque–Bera value illustrates that all the variables ($CO_2$, GDP, EC, TO, and AGRIC) do not comply with normality. Hence, the application of the linear techniques will yield misleading outcomes. Based on this, the current study used the wavelet approach to investigate the linkage between $CO_2$ and GDP, TO, AGRIC, and EC. We proceed to capture the stationarity features of variables of concern by utilizing traditional unit root tests (ADF and PP) and Zivot–Andrews (ZA) and Lee and Stractwich (LS) unit root tests proposed by Zivot and Andrews [57] and Lee and Strachwich [58], respectively. While the expectation of stationarityis not necessarily required when applying the wavelet approach [72,73]; its assumption offers a standard by which nonstationarity can be identified [67]. The outcomes of the traditional unit root test are depicted in Table 4 and the findings show that only AGRIC is stationary at level. Nonetheless, $CO_2$, TO, GDP, and EC are also found stationary after the first difference was taken. The outcomes of both ZA and LS, depicted in Table 5, also give credence to the outcomes of the ADF and PP unit root tests. After the stationarity feature of the series is confirmed, we can estimate the cointegration among the series using Bayer and Hanck's [64] combined cointegration test. The Bayer and Hanck [64] outcome is illustrated in Table 6, and findings show that $CO_2$, GDP, EC, TO, and AGRIC have a long-run relationship.

**Table 3.** Descriptive statistics.

|  | $CO_2$ | EC | GDP | TO | AGRIC |
|---|---|---|---|---|---|
| Mean | 0.8267 | 3134.5 | 815.5158 | 24.578 | $1.91 \times 10^{11}$ |
| Median | 0.7216 | 2790.4 | 595.0135 | 18.433 | $1.71 \times 10^{11}$ |
| Maximum | 1.9157 | 6923.9 | 2151.726 | 55.793 | $3.94 \times 10^{11}$ |
| Minimum | 0.3070 | 1234.19 | 345.42 | 7.6617 | $7.75 \times 10^{10}$ |
| Std. Dev. | 0.4876 | 1685.5 | 506.44 | 15.306 | $8.78 \times 10^{10}$ |
| Skewness | 0.8500 | 0.7644 | 1.1901 | 0.7157 | 0.654237 |
| Kurtosis | 2.5867 | 2.4315 | 3.3230 | 2.0704 | 2.363121 |
| Jarque–Bera | 7.0152 | 6.0972 | 13.223 | 6.6758 | 4.853106 |
| Probability | 0.0299 | 0.0474 | 0.0013 | 0.0355 | 0.088341 |
| Observations | 55 | 55 | 55 | 55 | 55 |

**Table 4.** Traditional unit root tests.

| | | ADF Unit Root Test | | |
|---|---|---|---|---|
| | | At Level I (0) | First Difference I (1) | Decision |
| | | T and I | T and I | |
| | GDP | −0.9012 | −6.4815 * | I (1) |
| | $CO_2$ | −2.6626 | −7.3821 * | I (1) |
| | EC | −2.5039 | −8.4014 * | I (1) |
| | TO | −1.3876 | −5.7691 * | I (1) |
| | AGRIC | −5.6106 * | −7.8427 * | I (0),I (0) |
| | | PP Unit Root Test | | |
| | GDP | −0.7133 | −9.8978 * | I (1) |
| | $CO_2$ | −2.6671 | −7.4401 * | I (1) |
| | EC | −2.4969 | −8.3571 * | I (1) |
| | TO | −1.8396 | −5.8967 * | I (0),I (1) |
| | AGRIC | −5.7122 * | −15.620 * | I (0) |

Note: 1% level of significance is illustrated by *.

**Table 5.** ZA and LS unit root test.

| | At Level I (0) | | First Difference I (1) | | Decision |
|---|---|---|---|---|---|
| | | ZA unit root test | | | |
| Variables | T and I | Break-Date | T and I | Break-Date | |
| GDP | −2.4908 | 1979 | −6.2685 ** | 1985 | I (1) |
| $CO_2$ | −2.9007 | 2000 | −8.2378 * | 1991 | I (1) |
| EC | −3.1018 | 1978 | −8.8344 * | 1991 | I (1) |
| TO | −3.8386 | 2004 | −6.9854 * | 1976 | I (1) |
| AGRIC | −7.0528 * | 1979 | −6.9761 * | 2002 | I (0), I (1) |
| | | LSunit root test | | | |
| GDP | −5.2403 | 1980 and 1997 | −8.8362 | 1977 and 1989 | I (1) |
| $CO_2$ | −4.6148 | 1984 and 1998 | −5.8828 *** | 1995 and 2004 | I (1) |
| EC | −4.8448 | 1992 and 2001 | −8.3239 | 1975 and 1978 | I (1) |
| TO | −5.6633 | 1991 and 2008 | −6.2901 ** | 1987 and 2001 | I (1) |
| AGRIC | −6.0759 ** | 1990 and 2002 | −7.9906 * | 1994 and 2009 | I (0), I (1) |

Note: 1%, 5% and 10% level of significance are illustrated by *, **, and *** respectively.

**Table 6.** Bayer–Hanch cointegration test.

| Model | Fisher Statistics | Fisher Statistics | Cointegration Decision |
|---|---|---|---|
| $CO_2$ = f(GDP, EC, TO, AGRIC) | EG-JOH | EG-JOH-BAN-BOS | |
| | 27.978 ** | 36.593 ** | Yes |
| | CV | CV | |
| 5% | 10.576 | 20.143 | |

Note: 5% significance level is depicted by **. EG, JOH, BAN, and BOS illustrate Engle–Granger, Johansen, Banerjee and Boswijk.

The current paper deployed the wavelet coherence (WTC) test to catch the correlation and causal linkage between $CO_2$ and AGRIC, EC, TO, and GDP in India between 1965 and 2019. This method is shaped from physics to obtain information that is previously unseen. Therefore, the research assesses the connection in the short-, medium-, and longrun between GDP and its regressors. Discussion is done inside the cone of influence (COI). The thick black contour illustrates a level of significance based on Monte Carlo simulations. Figure 4a–d, 0–4, 4–8, and 8–16 show short-, medium-, and longterm, correspondingly. Furthermore, the vertical and horizontal axis in Figures depicts frequency and time, respectively. Blue and yellow represent low and high dependence between the series. The rightward and leftward arrows illustrate positive and negative connections. Moreover, the right and down (leftward and up) illustrates that the first parameter leads (cause) the

second parameter, while the rightward and up (leftward and down) depict that the second parameter leads (cause) the first parameter. The findings of the WTC follow.

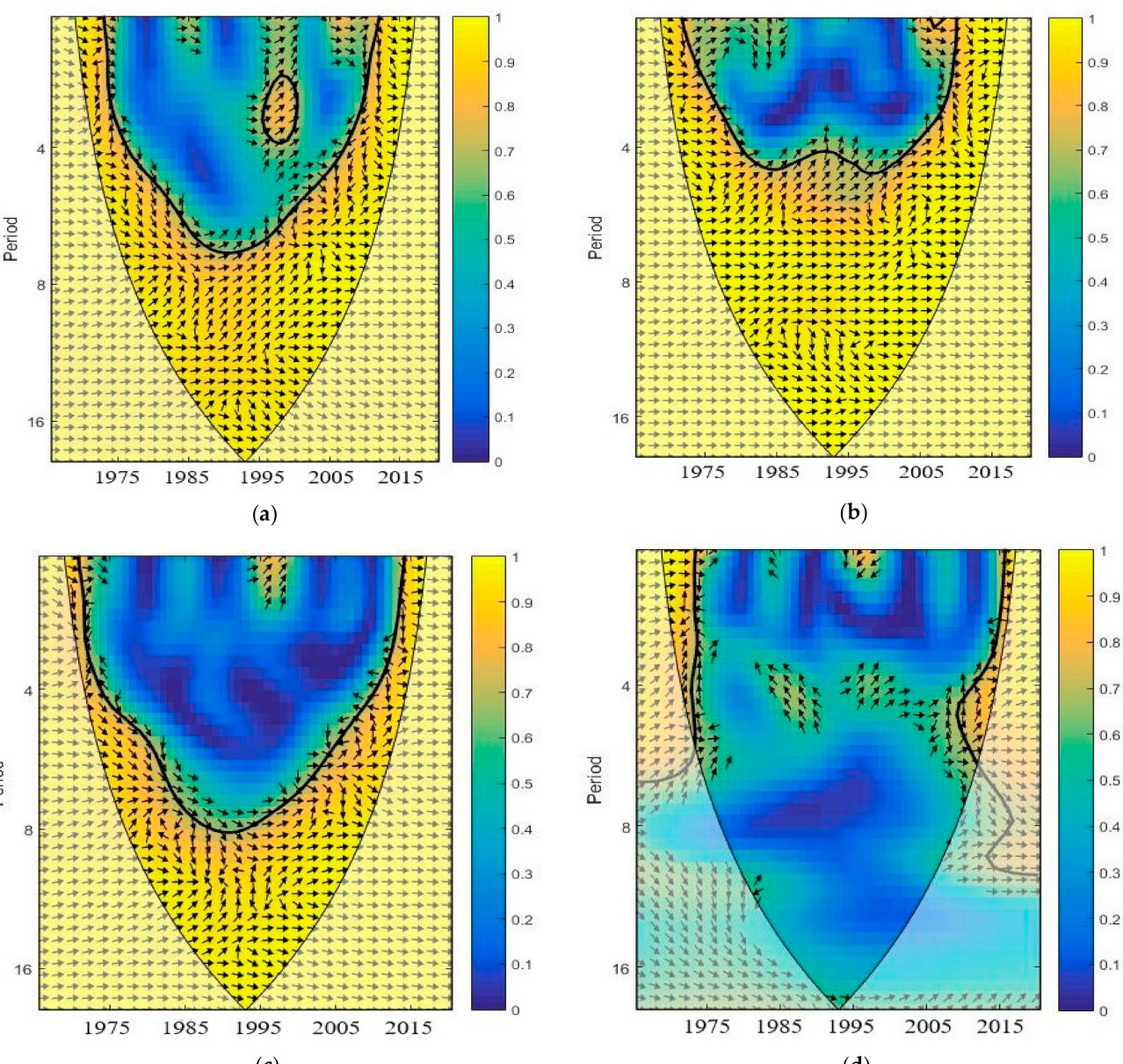

**Figure 4.** (**a**) WTC between $CO_2$ emissions and economic growth, (**b**) WTC between $CO_2$ emissions and energy consumption, (**c**) WTC between $CO_2$ emissions and agriculture, and (**d**) WTC between $CO_2$ emissions and trade Openness.

Figure 4a illustrates the WTC between GDP and $CO_2$ between 1965 and 2019. In the short term, the majority of the arrows are rightward, which illustrates evidence of a positive correlation between GDP and $CO_2$ emissions, although there is evidence of a correlation between $CO_2$ and GDP between 1975 and 2007. However, in the medium- and longterm between 1970 and 2019, the majority of the arrows are rightward, which illustrates an in-phase correlation between $CO_2$ and GDP in India. In summary, there is evidence of a positive correlation between GDP and $CO_2$ emissions in India between the periods of study, although it is more pronounced in the medium- and longterm. This implies that an increase in $CO_2$ emissions is accompanied by an upsurge in economic growth in India. This outcome implies that India's economic growth path is driven by $CO_2$ emission, which is astute, as the nation is ranked third highest emitter in the world. This outcome further

shows that India is still on the scale effect stage. This outcome validates the EKC hypothesis since an increase in economic growth is accompanied by an upsurge in $CO_2$ emissions. Our findings comply with the studies of Adebayo [19], Kirikkaleli et al. [52], Odugbesan and Adebayo [74], Khan et al. [17], Malik et al. [20], and Rjoub et al. [21].

Figure 4b shows energy consumption in India between 1965 and 2019. The majority of the arrows are rightward (positive correlation) in the short-run from the period 1965 to 1985 and from the period 2008 to 2019. However, in the medium- and longrun, the majority of the arrows are rightward, which shows that $CO_2$ and energy consumption are in-phase. Thus, an increase in energy consumption is followed by an increase in $CO_2$ emissions in India. The main motive for this in-phase, positive correlation between energy consumption and $CO_2$ emissions is that energy consumption from nonrenewable sources is high in India. Moreover, this outcome is not surprising since coal consumption is the nation's top energy source, accounting for 44% of the total energy use. Transitioning from nonrenewable to renewable energy sources takes time, technology, and a significant fixed cost. This is why producing energy from nuclear and natural gas is seen as a low-carbon alternative to energy produced from coal and oil [8–11]. Furthermore, adopting renewables is impossible without sufficient trained and technical manpower, which is a common issue in many emerging nations [14,19]. This outcome complies with the study of He et al. [23], Kalmaz and Adebayo [10], Zhang and Zhang [47], Olanrewaju et al. [8], Siddique et al. [27], Cheikh et al. [36], and Umar et al. [75], who established a positive connection between energy use and $CO_2$ emissions.

Figure 4c portrays the WTC between $CO_2$ emissions and agriculture in India between 1965 and 2019. The majority of the arrows are rightward, which illustrates in-phase relationship between $CO_2$ and agriculture in the short-run from period 1965 to 1976 and from the period 2012 to 2019.Nevertheless, in the medium- and longrun, most arrows are rightward, which shows that $CO_2$ and agriculture are in-phase. Thus, an increase in agriculture is accompanied by an upsurge in $CO_2$ emissions in India.This finding is expected since agriculture is a major source of greenhouse gases due to increased agricultural production volume, manure, livestock, crops, etc., which contribute to the greenhouse effect and climate change. According to the International Panel of Climate Change (IPCC), in 2013, agriculture, forestry, and the change of land use, account for as much as 25% of human-induced GHG emissions. Agriculture is one of the main sources of emitted methane and nitrous oxide. Our outcomes affirm Waheed's (2018) assertion that nitrous oxide and methane emissions from agricultural activities and land conservation are one of the major sources of $CO_2$ emissions in agriculture. In addition, the agricultural industry uses nonrenewable energy sources, including oil and diesel for irrigation, resulting in $CO_2$ emissions. As stated by Panhwar [76], farmers also use nitrogen-rich fertilizers to protect their crops. However, these fertilizers contribute to $CO_2$ emissions. Conventional farming practices should be replaced with modern approaches that serve to enhance productivity while lowering GHG emissions. This finding is consistent with the studies of Adebayo et al. [77] for South Korea, Waheed et al. [43] for Pakistan, Ben Jebli andBen Youssef [44] and for Brazil, and Dogan [39] for China.

Figure 4d shows the WTC between $CO_2$ emissions and trade openness in India between 1965 and 2019. In the short- and medium-term (high-frequency) from the period 1965 to 1975 and 2011 to 2019, the majority of the arrows are rightward (positive correlation) between $CO_2$ emission and trade openness. In the long run, however, there is little proof of a substantial association between $CO_2$ and trade openness. These mixed findings on the connection between trade openness and $CO_2$ can be translated as follows: a strong association between $CO_2$ emissions and trade openness is endorsed at low and medium scales until the mid-1980s, but then the association becomes less stable, eventually becoming insignificant in recent times. It may be claimed that the correlation between $CO_2$ emissions and trade openness is weak and cannot account for long-term patterns. This outcome complies with the findings of Mutascu [24] for France and Mahmoud et al. (2021) for Saudi Arabia, who disclosed a weak and positive correlation between $CO_2$ and

trade openness in the short- and medium-term, but found an insignificant correlation in the long-run. This, however, undermines the findings of Sebri and Ben-Salha [26], who found that international trade, would promote the transfer of green technologies, thereby assisting in the decarbonization of the power sector. It is possible to assume that TO has a very weak positive association with $CO_2$ since there is no proof of such a correlation much of the time. As a result, our contradictory observations do not affirm the presence of a stable $CO_2$–TO association in India. This outcome contradicts the findings of Sebri and Ben-Salha [35] for BRICS, Oh and Bhuyan [78] for Bangladesh, and Saidi and Mbarek [79] for 19 developing nations. The summary of the wavelet coherence outcomes is depicted in Table 7.

**Table 7.** Summary of the wavelet results.

| Frequency | Significance of the Correlation | Strength of the Correlation |
|:---:|:---:|:---:|
| High | $CO_2 \Leftrightarrow$ GDP (Yes) | Weak |
| Medium | $CO_2 \Leftrightarrow$ GDP (Yes) | Strong |
| Low | $CO_2 \Leftrightarrow$ GDP (Yes) | Strong |
| High | $CO_2 \Leftrightarrow$ EC (Yes) | Weak |
| Medium | $CO_2 \Leftrightarrow$ EC (Yes) | Strong |
| Low | $CO_2 \Leftrightarrow$ EC (Yes) | Strong |
| High | $CO_2 \Leftrightarrow$ TO (Yes) | Weak |
| Medium | $CO_2 \Leftrightarrow$ TO (Yes) | Weak |
| Low | $CO_2 \Leftrightarrow$ TO (No) | Null |
| High | $CO_2 \Leftrightarrow$ AGRIC (Yes) | Weak |
| Medium | $CO_2 \Leftrightarrow$ AGRIC (Yes) | Strong |
| Low | $CO_2 \Leftrightarrow$ AGRIC (Yes) | Strong |

Notes: $\Leftrightarrow$ illustrates the relationship, and GDP, AGRIC, EC, and $CO_2$ depict economic growth, agriculture, energy use, and $CO_2$ pollution.

Table 8 illustrates the outcomes of the Gradual shift causality. The advantage of the Gradual shift causality test is that it can catch causal linkage between series in the presence of break(s) in series. We see that the causality outcomes confirm that $CO_2$ emissions Granger causes GDP in India, which illustrates that $CO_2$ emission can predict significant variation in economic growth. This result is consistent with the findings of Adebayo and Kirikkaleli [13] for Japan, Zhang et al. [14] for Malaysia, He et al. [23] for Mexico, and Akinsola and Adebayo [25] for Thailand. In addition, at a significance level of 1%, there is evidence of unidirectional causality from energy consumption to $CO_2$ emissions. This infers that significant variation in $CO_2$ emissions can be predicted by energy consumption. This outcome complies with the studies of Olanrewaju et al. [8] for Indonesia and Rjoub et al. [21] for Turkey. Lastly, at a significance level of 1%, there is evidence of two-way causality between$CO_2$ emissions and agriculture, signifying that both $CO_2$ emissions and agriculture can predict each other. This outcome concurs with the study of Waheed et al. [43] for Pakistan. The findings from the Gradual shift causality test have significant implication for policymakers in Pakistan. Additionally, the Gradual shift causality test outcomes provide supportive evidence for the wavelet coherence test outcomes.

**Table 8.** Gradual shift causality test.

| Causality Path | WaldStat | No. of Fourier | *p*-Value | Decision |
|---|---|---|---|---|
| GDP $\rightarrow$ CO$_2$ | 4.3821 | 3 | 0.7348 | Do not Reject Ho |
| CO$_2 \rightarrow$ GDP | 14.031 *** | 3 | 0.0505 | Reject Ho |
| EC $\rightarrow$ CO$_2$ | 27.609 * | 3 | 0.0002 | Reject Ho |
| CO$_2 \rightarrow$ EC | 6.8497 | 3 | 0.4446 | Do not Reject Ho |
| AGRIC $\rightarrow$ CO$_2$ | 25.7567 * | 2 | 0.0000 | Reject Ho |
| CO$_2 \rightarrow$ AGRIC | 27.131 * | 2 | 0.0000 | Reject Ho |
| TO $\rightarrow$ CO$_2$ | 10.050 | 2 | 0.1857 | Do not Reject Ho |
| CO$_2 \rightarrow$ TO | 5.4607 | 2 | 0.6039 | Do not Reject Ho |

Note: 1%, and 10% levels of significance are illustrated by *, and ***, respectively.

## 5. Conclusions and Policy Direction

The present study assesses the interconnection between environmental degradation and agriculture taking into account the role of economic growth, energy consumption, and trade openness in India between 1965 and 2019. No prior studies have assessed this interconnection using the novel wavelet coherence approach, to the best of the investigators' understanding. To achieve the research objectives, the study utilized both wavelet coherence and Gradual shift causality tests. The novelty behind wavelet coherence is that it can decompose time series into different time scales and therefore illustrates the connection between parameters. On the other hand, simply analyzing the data with linear techniques may provide misleading results, as this could hide information that might influence the observed relationships. Although this empirical strategy has not been applied to this topic so far, it brings consistent correlating evidence with far-reaching policy implications for India. Finally, to provide evidence of causal inferences among the variables, the present study utilized the Gradual shift causality test. The main innovation behind this test is that it can capture causality between series in the presence of a structural break(s). The findings from the wavelet coherence test revealed (a) a strong positive correlation between CO$_2$ emissions and GDP in the medium- and longterm, (b) a strong positive correlation between CO$_2$ emissions and agriculture predominantly in the medium- and longterm, (c) a significant and positive correlation between agriculture and CO$_2$ emissions in the medium- and longterm, and (d) a weak and positive correlation between trade openness and CO$_2$ emissions in the medium term. In summary, there is a positive correlation between CO$_2$ emissions and agriculture, trade openness, and energy use, predominantly in the medium- and longterm. This suggests that an upsurge in CO$_2$ emissions and agriculture, trade openness, and energy use in India decrease environmental sustainability. Furthermore, the Gradual shift causality test outcomes revealed a one-way causality from energy consumption and economic growth to CO$_2$ emissions, while there is feedback causality between agriculture and emissions.

Based on the findings, the following policy suggestions are formulated. First, at the national level, the government of India should be careful when formulating economic expansion policies that will jeopardize environmental sustainability. Second, the total energy mix should be changed by substituting nonrenewable energy sources with green energy sources, including solar, wind, and hydro. At the regional and local levels, the Indian government should allow private businesses to invest in green energy use, production, and innovation to achieve this aim. Third, the Indian government needs to initiate agricultural reforms, such as the implementation of the National Agricultural Policy. To decrease CO$_2$ emissions from agricultural production, small farmers should utilize solar irrigation pumps, organic farming, and tunnel farming. Finally, tree planting is an effective method of reducing CO$_2$ emissions. To minimize CO$_2$ emissions, the Indian government should take measures regarding afforestation and reforestation, including the "Billion Tree Tsunami" project and monitor deforestation. It is known that enhancing trade flows increases the consumption of energy (mostly fossil fuels for transport and industry purposes) and pollutants; therefore, policies should target the development of green practices along

the supply chain in India, with a specific focus on the establishment of low-carbon production activities. Innovation could also play a valuable role. This could not only reduce the environmental externalities but also boost long-term business profitability. Finally, increased dependence on green energy solutions and moving away from fossil fuel-based energy solutions will aid economic development patterns in mitigating $CO_2$ emissions, which will have a beneficial effect on the environment. This will support India in making strides toward achieving the SDG 13 targets. Although the present study used a novel technique to investigate this association, it only used $CO_2$ emissions as proxy of environmental degradation. Thus, other studies should use other proxies of environmental degradation to investigate this association. Further studies should be conducted on developing and developed countries using other determinants of $CO_2$ emission that were not investigated in this empirical analysis.

**Author Contributions:** Conceptualization, T.S.A. and D.K.; methodology, D.K. and T.S.A.; software, T.S.A. and S.Y.G.; validation, A.O.; formal analysis, A.O. and D.K.; investigation; T.S.A. and D.K.; resources, T.S.A.; data curation, S.Y.G.; writing—original draft preparation, D.K., T.S.A. and S.Y.G.; writing—review and editing, A.O. and D.K.; visualization, D.K.; supervision, A.O. and S.Y.G.; project administration, T.S.A. and D.K. All authors have read and agreed to the published version of the manuscript.

**Funding:** This research received no external funding.

**Institutional Review Board Statement:** The study was conducted according to the research guidelines approved by the Ethics Committees of Authors Institutions.

**Informed Consent Statement:** Not applicable.

**Data Availability Statement:** Not applicable.

**Conflicts of Interest:** The authors declare no conflict of interest.

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
