# Peer review of "Investigating the Linkage between Economic Growth and Environmental Sustainability in India: Do Agriculture and Trade Openness Matter?"

_sustainability, doi:10.3390/su13094753_

Round 1

Reviewer 1 Report

Dear authors, 

you have put quite some effort in your paper it is quite long and well established. 

the situation in India and the consumption of CO2 might be a topic to write a book which forwarded me to the following: each part of India is different so cross-country or regional approach would reveal more insight in your theme. 

The logical rise of GDP is not only a consequence of economy and in it agriculture so the population would be a good variable to use in analysis

in table 1 about the literature, you might input some real figures to see what other authors found

Also their findings and their figures of relation should be compared in your discussion and conclusions

Readers would then understand the real value of your findings

what boders me is the situation that agriculture is not only a producer of CO2 but also a producer of O2 that was not at all mentioned in the text

also on page 15 only manure and fertilizers do not cause Co2 rise

what about the livestock? their existence are most influential for Co2

again the rise of population and subsistence agriculture was not mentioned

supposing the paper might be improved I wish you success in publishing

Author Response

Reviewer 1

Dear authors, 

you have put quite some effort in your paper it is quite long and well established. 

the situation in India and the consumption of CO2 might be a topic to write a book which forwarded me to the following: each part of India is different so cross-country or regional approach would reveal more insight in your theme. 

Thank you for this suggestion. However, this paper only focused on India as a whole not region

The logical rise of GDP is not only a consequence of economy and in it agriculture so the population would be a good variable to use in analysis

Thank you for this suggestion. Yes urbanization is an important determinant of CO2. Nonetheless, the study only focused on the variables utilized. This is one of the area we highlighted that future studies can conduct research on.

in table 1 about the literature, you might input some real figures to see what other authors found

Thank you for this suggestion. We summary the literature and also utilise table 1 to reflects the findings of prior studies and as well use (+), (-),ðand ð to establish the type of relationship and causality. This is done to reflect clear findings from previous studies.

Also their findings and their figures of relation should be compared in your discussion and conclusions

Thank you for this suggestion. As suggested our findings were compared to prior studies in the discussion part

Readers would then understand the real value of your findings

Thank you for this suggestion. As suggested the corrections were incorporated

what boders me is the situation that agriculture is not only a producer of CO2 but also a producer of O2 that was not at all mentioned in the text

Thank you for this suggestion. Yes agriculture not only a producer of CO2 but also a producer of O2. This study only focused on the effect of agricultural activities on CO2 emissions

on page 15 only manure and fertilizers do not cause Co2 rise. what about the livestock? their existence are most influential for Co2

Thank you for these suggestions. As suggested other agricultural activities that leads to rise in CO2 emissions are incorporated

again the rise of population and subsistence agriculture was not mentioned

Thank you for this suggestion. Though, urbanisation is one of the the determinants of CO2 emissions, this study only examined the effect of energy use, trade openness, economic growth and agriculture on CO2 emissions.

Reviewer 2 Report

There are some remarks for the improvement of the article:

  • Giving the abbreviations for the 1st time – it should be given / or considered to give full words (in the text we see abbreviations such as: ARDL, VECM, FMOLS, DOLS, OLS, CCR, GMM, and PVAR and some places in further text – 2.1.1. Subchapter)
  • Why the period of 1965–2019 years was considered to use in the analysis? In abstract we see 1965–2019 years period, but in the part 3.1. Data – there was mention” utilizing data stretching between 1970 and 2019 for all indicators“. So why do these year differ?
  • In part 2. Literature Review the sentence „The theoretical framework of the study discusses the environmental Kuznets curve (EKC) theory“ – seems to be not finished.
  • In some places grammar should be revised.
  • Maybe it is better to give Figure 3: Analysis Flow Chart before the Figure 2, as the Figure 3 represents all steps of implemented analysis.
  • In the part 5. Conclusion and Policy Direction, we see recommendations for policy makers – but to whom it may concern directly – maybe it can be detailed – national, regional or local level?
  • What were the limitations of the research?

Author Response

Reviewer 2

There are some remarks for the improvement of the article:

  • Giving the abbreviations for the 1st time – it should be given / or considered to give full words (in the text we see abbreviations such as: ARDL, VECM, FMOLS, DOLS, OLS, CCR, GMM, and PVAR and some places in further text – 2.1.1. Subchapter)
  • Thank you for this suggestion the full meaning of the abbreviation were incorporated
  • Why the period of 1965–2019 years was considered to use in the analysis? In abstract we see 1965–2019 years period, but in the part 3.1. Data – there was mention” utilizing data stretching between 1970 and 2019 for all indicators“. So why do these year differ?

Thank you for this suggestion. It was a mistake which was corrected in the revised version. The study period was between 1965 and 2019.

  • In part 2. Literature Review the sentence „The theoretical framework of the study discusses the environmental Kuznets curve (EKC) theory“ – seems to be not finished.
  • Thank you for this suggestion. As suggested, the EKC hypothesis was improved
  • In some places grammar should be revised.
  • Thank you for this suggestion. As suggested, the manuscript was proofread by a native speaker
  • Maybe it is better to give Figure 3: Analysis Flow Chart before the Figure 2, as the Figure 3 represents all steps of implemented analysis.
  • Thank you for this suggestion. As suggested this was changed
  • In the part 5. Conclusion and Policy Direction, we see recommendations for policy makers – but to whom it may concern directly – maybe it can be detailed – national, regional or local level?
  • Thank you for this suggestion. As suggested, the conclusion was improved
  • What were the limitations of the research?
  • Thank you for the suggestions as suggested limitations of the research was incorporated.

Reviewer 3 Report

Referee Report

sustainability-1172033

“Investigating the Linkage between Economic Growth and Environmental Sustainability in India: Does Agriculture and Trade Openness Matter?”

This paper could be interesting for the researchers in “open economy macroeconomics and development economics”.  The paper simply investigates the relationship between CO2 emissions and economic growth in India.  Author(s) use a time-series data for the years between 1965 – 2019.  Their results show most variables appear highly correlated with CO2 emissions.  The present study is an empirical study that supports all the arguments presented and adequately convincing.

My comments:

  • The present paper offers clear evidence on environmental sustainability in India. The paper displays adequate understanding of the related literature in the field and cites an appropriate range of literature sources with a strong analysis.  Author(s) also constructed their arguments on a correct base of theory.  The methodology was correctly applied, and the implications are well articulated.  Provided results overlap with previous research. 
  • The current study has practical knowledge and contains academic contribution in the open economy macroeconomics field for the Sustainability.

Author Response

Reviewer 3

Referee Report

sustainability-1172033

“Investigating the Linkage between Economic Growth and Environmental Sustainability in India: Does Agriculture and Trade Openness Matter?”

This paper could be interesting for the researchers in “open economy macroeconomics and development economics”.  The paper simply investigates the relationship between CO2 emissions and economic growth in India.  Author(s) use a time-series data for the years between 1965 – 2019.  Their results show most variables appear highly correlated with CO2 emissions.  The present study is an empirical study that supports all the arguments presented and adequately convincing.

 My comments:

  • The present paper offers clear evidence on environmental sustainability in India. The paper displays adequate understanding of the related literature in the field and cites an appropriate range of literature sources with a strong analysis.  Author(s) also constructed their arguments on a correct base of theory.  The methodology was correctly applied, and the implications are well articulated.  Provided results overlap with previous research. 
  • The current study has practical knowledge and contains academic contribution in the open economy macroeconomics field for the Sustainability.

Dear Reviewer:

Thank you for the comments
